# Advances in Autophagy–Lysosomal Pathway and Neurodegeneration via Brain–Gut Axis

**DOI:** 10.3390/biomedicines13061390

**Published:** 2025-06-05

**Authors:** Ping Yao, Hailong Han

**Affiliations:** Institute of Cytology and Genetics, School of Basic Medical Sciences, Hengyang Medical School, University of South China, Hengyang 421001, China; 20222013111202@stu.usc.edu.cn

**Keywords:** lysosome, autophagy, gut–brain axis, neurodegenerative diseases, gut microbiota

## Abstract

**Background/Objectives:** The autophagy–lysosomal pathway (ALP) is crucial for neuronal health by clearing misfolded proteins and damaged organelles. While much research has focused on ALP dysfunction in the central nervous system, new evidence shows its importance in the gut, where it affects neurodegeneration via the gut–brain axis. Past reviews have mainly studied the ALP’s direct neuroprotective effects or the gut microbiota’s role in neurodegeneration separately. However, the two-way relationship between the ALP and the gut microbiota in neurodegenerative diseases is not well understood. We combine the latest findings on the ALP’s role in gut health, microbial imbalance, and neuroinflammation, providing a comprehensive view of their combined effects in Alzheimer’s, Parkinson’s, and Huntington’s diseases. **Methods:** This narrative review synthesizes evidence from preclinical, clinical, and translational studies (2014–2025) to explore the interplay between the autophagy–lysosomal pathway (ALP) and the gut–brain axis in neurodegeneration. The literature was identified via PubMed and Web of Science using search terms including autophagy, lysosome, gut microbiota, neurodegeneration, and gut–brain axis, with additional manual screening of reference lists. The inclusion criteria prioritized studies elucidating molecular mechanisms (e.g., ALP–microbiota crosstalk), while excluding case reports or non-peer-reviewed sources. **Results:** The gut–brain axis facilitates bidirectional communication between the gut and the brain through neural, immune, and metabolic pathways. Autophagy dysfunction may disrupt intestinal homeostasis, promote gut microbiota dysbiosis, and trigger chronic neuroinflammation, ultimately accelerating neurodegeneration. Notably, strategies targeting the gut microbiota and restoring intestinal barrier function via the ALP have demonstrated promising potential in delaying the progression of neurodegenerative diseases. **Conclusions:** This review establishes the ALP as a dynamic regulator of gut–brain communication, highlighting microbiota-targeted therapies as promising strategies for neurodegeneration.

## 1. Introduction

Neurodegenerative diseases (NDDs) are a group of neurological disorders characterized by progressive neuronal loss in the central nervous system (CNS) or peripheral nervous system (PNS), primarily including Alzheimer’s disease (AD), Parkinson’s disease (PD), and Huntington’s disease (HD) [1]. According to the World Health Organization’s Global Dementia Observatory, as of 2021, approximately 55.2 million people worldwide were living with dementia, with Alzheimer’s disease being the most common form. This number is projected to nearly triple to 152 million by 2050 due to population aging and growth [2]. Historically, research has focused on central nervous system (CNS)-centric mechanisms, such as amyloid-β (Aβ) plaques in AD and α-synuclein (α-syn) aggregation in PD. However, emerging evidence highlights the gut–brain axis (GBA) as a critical mediator of neurodegeneration, with bidirectional communication involving neural, immune, and metabolic pathways [3]. Clinical evidence demonstrates that over 80% of PD patients exhibit gastrointestinal dysfunction such as constipation, salivary secretion abnormalities, defecation disorders, nausea, and dysphagia, with approximately half developing constipation prior to the onset of motor symptoms [4,5], suggesting that GBA dysregulation plays a critical role in the pathogenesis of NDDs and may serve as a potential target for early intervention.

Intestinal homeostasis relies on a dynamic equilibrium among the gut microbiota, intestinal epithelial barriers, and immune system. Disruption of this homeostasis can exert systemic neurological effects via the GBA. Research indicates that gut dysbiosis may exacerbate neurodegenerative progression through multiple pathways, including vagus nerve-mediated signaling, metabolite-induced neuroendocrine imbalance, and immune-inflammatory cascades [6,7]. Consequently, maintaining intestinal homeostasis is crucial for NDD management.

The autophagy–lysosomal pathway (ALP), a critical catabolic process utilizing lysosomes for cellular component degradation, plays an essential role in preserving intestinal homeostasis. Dysfunction of this pathway, characterized by abnormal protein aggregation and impaired organelle accumulation in neurons, represents a pathological hallmark of various NDDs [8]. While previous reviews have established the ALP’s role in neuronal homeostasis, recent studies reveal its regulatory influence on the peripheral organs, particularly the gut. For instance, intestinal epithelial cell-specific Atg5 or Atg7 knockout disrupts the gut microbiota balance, enriching pro-inflammatory bacteria (e.g., *Pasteurella*) and exacerbating systemic inflammation via disrupted barrier integrity [9]. Conversely, gut microbiota-derived metabolites, such as butyrate and indole-3-lactic acid, enhance ALP activity through the AMPK/mTOR and AhR-TFEB pathways, respectively [10,11]. These findings position the ALP as a bidirectional modulator of gut–brain crosstalk, yet comprehensive reviews integrating the ALP’s peripheral and central roles in NDDs remain scarce.

By bridging the mechanistic insights of autophagy regulation with the emerging concept of the GBA, this review provides a comprehensive perspective on how targeting the ALP—both centrally and peripherally—could offer novel therapeutic approaches for NDDs. It further highlights translational innovations, such as fecal microbiota transplantation (FMT) and drug discovery via autophagy tethering, which target ALP dysregulation across organs [12,13]. These advancements not only refine existing hypotheses but also pave the way for multi-modal therapies addressing the systemic nature of neurodegeneration.

## 2. Intestinal Homeostasis and Neurodegenerative Diseases

### 2.1. Alzheimer’s Disease

The main pathological features of Alzheimer’s disease (AD) are senile plaques formed by β-amyloid protein (Aβ) deposits and neurofibrillary tangles composed of hyperphosphorylated tau protein [14]. Clinical studies show that AD patients have significant changes in their gut microbiota. For example, beneficial bacteria like *Bifidobacterium* and *Lactobacillus* decrease, while harmful bacteria like *Escherichia coli* and *Helicobacter pylori* increase [15]. A recent study also found that AD patients have a less diverse gut microbiota, with more pro-inflammatory bacteria [16].

In 2020, Kim et al. reported that fecal microbiota transplantation (FMT) from wild-type donor mice into ADLPAPT transgenic mice (a model of AD-like pathology with amyloid plaques and neurofibrillary tangles) effectively alleviated AD-related pathological features, including Aβ plaque deposition, tau pathology, and cognitive impairment [17]. Similarly, studies using the 3xTg-AD mouse model have shown that mice housed under specific pathogen-free (SPF) conditions exhibited more severe AD phenotypes compared to germ-free (GF) mice. Moreover, FMT from AD patients into GF 3xTg-AD mice significantly ameliorated AD-related symptoms [16]. Both germ-free conditions and antibiotic treatment significantly reduced Aβ plaque deposition in the brains of APP/PS1 transgenic mice, further supporting the critical role of the gut microbiota in AD pathology [18,19].

### 2.2. Parkinson’s Disease

The neuropathological features of Parkinson’s disease (PD) are primarily characterized by the loss of dopaminergic neurons in the substantia nigra of the brain and the intracellular aggregation of the α-synuclein (α-syn) protein [20]. Clinical data show that middle-aged and elderly PD patients have significant changes in their gut microbiota compared to healthy people [21].

Animal model studies have further demonstrated that the development of α-syn pathology, microglial activation, and motor deficits in Thy1-αSyn (α-syn-overexpressing) mice are influenced by the gut microbiota. Thy1-αSyn mice housed under specific pathogen-free (SPF) conditions exhibited more severe PD-like symptoms than those raised under germ-free or antibiotic-treated conditions [7]. Interestingly, fecal microbiota transplantation (FMT) from PD patients into germ-free mice aggravated motor dysfunction compared to FMT from healthy controls. Moreover, FMT from healthy donors significantly alleviated α-syn-mediated motor impairments in Thy1-αSyn mice [7].

Similarly, transgenic rat models overexpressing human α-syn demonstrated age-dependent gut dysbiosis, and short-term antibiotic treatment alleviated the α-syn expression in the forebrain [22]. When gut bacteria from MPTP (a neurotoxin)-treated PD model mice were transplanted into healthy mice, the healthy mice developed movement problems and neurotransmitter loss. Conversely, transplants from healthy mice improved gut inflammation, brain cell activation, neurotransmitter issues, and movement problems in MPTP model mice [23]. Chronic rotenone administration induced gastrointestinal dysfunction and motor symptoms in conventionally housed mice, but not in germ-free mice, highlighting the critical role of the gut microbiota in PD pathogenesis [24].

### 2.3. Huntington’s Disease

Huntington’s disease (HD) is an autosomal dominant neurodegenerative disorder caused by the abnormal expansion of CAG trinucleotide repeats in the Huntingtin (HTT) gene. This mutation results in the misfolding and accumulation of mutant huntingtin protein within various neural cell types, including neurons, microglia, and astrocytes [25]. In addition to motor, cognitive, and psychiatric disturbances, HD patients often suffer from gastrointestinal dysfunction, such as malnutrition, diarrhea, and unintended weight loss [26,27].

Although studies exploring the relationship between HD and the gut microbiota are limited, recent research has reported significant alterations in the gut microbiota composition in HD patients compared to healthy controls [28]. Wasser et al. found that male HD gene carriers had lower levels of Firmicutes, Lachnospiraceae, and Akkermansiaceae compared to healthy people. Akkermansiaceae (e.g., *Akkermansia muciniphila*) help maintain the gut barrier, while Lachnospiraceae (e.g., *Blautia* spp. And *Coprococcus* spp.) and Firmicutes (e.g., *Faecalibacterium prausnitzii* and *Roseburia* spp.) produce butyrate, which reduces inflammation [28].

Similar alterations in the gut microbiota composition have been observed in HTT R6/1 transgenic HD mouse models compared with wild-type (WT) mice. R6/1 mice exhibited motor deficits and weight loss, accompanied by significant gut microbiota dysbiosis [29]. Additionally, studies in R6/2 transgenic mice reported gut microbiota imbalance and intestinal barrier damage, characterized by a shorter colon length, increased intestinal permeability, a reduced body weight, and a smaller body size compared with age-matched WT mice [30]. Both R6/1 and R6/2 HD mice exhibited increased *Bacteroides* spp. (pro-inflammatory) and decreased *Roseburia*/*Faecalibacterium* spp. (butyrate-producing) versus WT mice [29,30].

## 3. Gut–Brain Axis Mechanisms in Neurodegenerative Diseases

The bidirectional communication between the gut and the brain is primarily mediated through three interconnected pathways: the nervous system, microbial metabolites, and the immune system [31]. These pathways involve microbial metabolites, peptides, gut hormones, neurotransmitters, inflammatory factors, and immune cells (Figure 1). Changes in gut function travel to the brain through the ENS-CNS, circulatory system, or immune system and affect brain activity [32]. In turn, the CNS can regulate gastrointestinal motility and homeostasis through similar mechanisms, highlighting the dynamic interplay between the gut and brain [33]. These pathways contribute to the modulatory effects of the gut–brain axis (GBA) in the progression of neurodegenerative diseases.

### 3.1. Nervous System

The ENS and CNS form a bidirectional regulatory network within the GBA, playing a pivotal role in the pathophysiological processes of neurodegenerative diseases. The ENS, a complex intrinsic neuronal network, is predominantly distributed within the mucosal and muscular layers of the gastrointestinal tract, orchestrating diverse digestive functions [34]. It contains many neurons and glial cells, forming two main nerve clusters: the myenteric plexus and submucosal plexus [35]. These plexuses coordinate gastrointestinal motility and secretory functions through the modulation of smooth muscle contractions and glandular activity [36].

One of the critical mechanisms by which the gut microbiota regulates ENS function is through the activation of pattern recognition receptors (PRRs), such as Toll-like receptors (TLRs), which detect microbial-derived molecules [37]. Notably, studies have demonstrated that Tlr2-deficient mice exhibit impaired small intestinal motility. Co-culture experiments using smooth muscle cells isolated from wild-type mouse ilea partially restored the morphological and functional abnormalities of enteric neurons in Tlr2-deficient mice. This neuroprotective effect is closely associated with the enhanced production of neurotrophic factors, including glial cell-derived neurotrophic factor (GDNF), within intestinal smooth muscle cells [38]. Another study found that mice lacking Tlr4 produced fewer and drier fecal pellets, indicating changes in nerve plexus function [38].

The microbial environment exerts profound effects on neurotransmitter levels within the host. For instance, GF mice display significantly altered neurotransmitter metabolism compared to conventionally raised mice. In particular, GF mice exhibit elevated turnover rates of norepinephrine, dopamine, and serotonin in the striatum, accompanied by markedly reduced mRNA expression of nerve growth factor-induced protein A (NGFI-A) in the prefrontal cortex and brain-derived neurotrophic factor (BDNF) in the hippocampus and amygdala [39]. Intriguingly, cross-colonization experiments further support the critical role of the gut microbiota in regulating host neurobiology. The colonization of BALB/c GF mice with the gut microbiota from NIH Swiss mice enhanced their exploratory behavior, concomitant with increased hippocampal BDNF levels [40].

### 3.2. Metabolites

Gut microbes produce metabolites like short-chain fatty acids (SCFAs), branched-chain fatty acids, bile acid derivatives, and vitamins, which play key roles in the body. The most studied are SCFAs (e.g., butyrate, acetate, propionate) derived from the bacterial fermentation of dietary fiber [41]. Accumulating evidence indicates that butyrate exerts critical regulatory effects on energy homeostasis, immune modulation, colonic motility, and intestinal inflammation [42,43,44]. Interestingly, recent studies have revealed that SCFAs also exhibit important functions within the central nervous system (CNS) [41]. For instance, the oral administration of *Clostridium butyricum* has been shown to enhance the antioxidant and anti-inflammatory capacities of mice subjected to cerebral ischemia–reperfusion injury, thereby protecting the nervous system from reperfusion-induced damage. This neuroprotective effect was closely associated with increased butyrate levels in the brain [45].

SCFAs have been reported to modulate neuroinflammation by regulating the activation and proliferation of regulatory T cells (Tregs). Mechanistically, SCFAs influence the histone modification of the Foxp3 (Forkhead box protein 3) promoter region, a key transcriptional regulator of Tregs, thereby enhancing their immunomodulatory function. Moreover, long-chain fatty acids (LCFAs) have also been implicated in the regulation of neuroimmune responses. For example, lauric acid (C12 FA), also known as dodecanoic acid, has been shown to promote the differentiation and proliferation of helper T cells (Th1 and/or Th17 cells) via the p38 mitogen-activated protein kinase (p38-MAPK) signaling pathway in experimental autoimmune encephalomyelitis (EAE) mouse models. Notably, lauric acid treatment reduced the retention of Th1/Th17 cells in the gut, ultimately increasing the severity of EAE symptoms [46].

In addition to fatty acids, the gut microbiota can synthesize a variety of neuroactive metabolites, including histamine, dopamine, norepinephrine, acetylcholine, γ-aminobutyric acid (GABA), and melatonin [47]. These microbial-derived neurotransmitters can directly influence neuronal activity by entering the circulatory system and acting on neurons within the brain [48]. Gut-derived hormones such as cholecystokinin, ghrelin, and serotonin have also been implicated in the regulation of mood disorders, including depression and anxiety (Figure 2).

### 3.3. Immune System

The disruption of intestinal homeostasis can trigger a cascade of immune responses, which may amplify chronic inflammation and contribute to the development of neurodegenerative diseases [49]. The dissemination of pro-inflammatory cytokines into the brain has been shown to induce protein misfolding and aggregation within neurons, axonal damage, and demyelination, ultimately promoting neurodegeneration [50,51]. Notably, increasing evidence suggests a close association between gut microbiota dysbiosis and the rising incidence of AD [52].

Gut microbiota imbalance may compromise intestinal barrier integrity, leading to increased gut permeability and the translocation of bacteria or endotoxins across the epithelial barrier. This, in turn, elicits immune activation and the release of pro-inflammatory cytokines, which can impair the blood–brain barrier (BBB) and adversely affect neuronal function [53]. Gut bacteria imbalance also affects the brain via immune cells. Compounds from gut bacteria activate TREM receptors on macrophages, boosting inflammation and damaging the gut barrier. Activated macrophages and inflammatory factors then travel through the bloodstream to harm the brain [54].

The gut microbiota has also been implicated in the regulation of glial cell function, thereby indirectly modulating neuronal activity. Erny et al. reported that germ-free mice exhibited functionally impaired microglia with reduced immune responsiveness compared to conventionally colonized mice, highlighting the essential role of the microbiota in maintaining microglial function [6]. Furthermore, Barroso et al. demonstrated that the aryl hydrocarbon receptor (AhR), expressed on astrocytes, can be activated by microbiota-derived tryptophan metabolites such as indole-3-aldehyde (IAld) and indoxyl-3-sulfate (I3S). The activation of AhR subsequently triggers the type I interferon signaling pathway, promoting the expression of anti-inflammatory cytokines such as interleukin-10 (IL-10), while suppressing pro-inflammatory factors including interleukin-6 (IL-6) and tumor necrosis factor-alpha (TNF-α), thus exerting neuroprotective anti-inflammatory effects [55].

## 4. Role of Autophagy in Gut–Brain Axis Regulation of Neurodegenerative Diseases

### 4.1. Autophagy–Lysosomal Pathway

The autophagy–lysosomal pathway (ALP) is a vital intracellular degradation system responsible for removing damaged organelles and aggregated proteins, thereby maintaining cellular homeostasis. It consists of a series of orchestrated steps: initiation, autophagosome formation, maturation, fusion with lysosomes, and cargo degradation. These processes are regulated by autophagy-related genes (ATGs) and modulated by key signaling molecules such as mTORC1, AMPK, TFEB, and lysosomal components like ATP13A2 and Rab GTPases [56]. As a key homeostatic mechanism, the ALP has been implicated in a variety of physiological and pathological processes, including neurodegenerative diseases, infectious diseases, cancer, and aging [57]. Recent studies have provided new insights into the role of the ALP in neurodegeneration. For example, impaired lysosomal acidification in astrocytes has been linked to neuroinflammation and Alzheimer’s disease progression, while enhanced astrocytic autophagy improves amyloid-β clearance and cognitive function [58,59]. These findings suggest that restoring ALP function in specific cell types holds therapeutic potential.

Moreover, ALP dysregulation is not confined to the central nervous system. Emerging evidence suggests that peripheral autophagy, particularly in the gut, influences neurodegenerative processes through the gut–brain axis. The disruption of intestinal autophagy affects the microbial balance and epithelial integrity, contributing to systemic inflammation and brain dysfunction [60]. Microbial modulation, such as through probiotics, has been shown to activate autophagy-related pathways and support mucosal and neural health [61]. Additionally, the identification of selective autophagy mechanisms such as aggrephagy opens up new avenues for targeted therapies. The newly discovered receptor CCT2, for instance, facilitates the clearance of solid mutant huntingtin aggregates, offering a promising strategy for Huntington’s disease treatment [13]. These advances highlight the ALP as a versatile and increasingly druggable target in neurodegenerative diseases (Table 1).

### 4.2. Autophagy and Gut Homeostasis

#### 4.2.1. Autophagy in Gut Cell Function

Autophagy plays a crucial role in maintaining epithelial cell homeostasis and protecting against infection and inflammation, thereby preserving intestinal barrier integrity under stress conditions. It achieves this by regulating tight junctions and preventing cell death, which are essential for gut barrier function during stress [70]. Autophagy dysfunction can affect various types of intestinal cells, such as Paneth cells (PCs) and intestinal stem cells (ISCs), leading to gut homeostasis imbalance and inflammation. PCs are specialized gut cells in the small intestine that release antimicrobial peptides to defend against infections and maintain the gut environment [71]. Cadwell et al. discovered that Crohn’s disease patients with a mutation in the autophagy-related gene Atg16l1 (T300A) exhibited abnormalities in the PC morphology, reduced granule secretion, and a diffuse distribution of lysozyme. These findings suggest that autophagy in PCs is essential in Crohn’s disease [72]. Further studies on mice with the specific knockout of autophagy genes Atg16l1, Atg5, and Atg7 in intestinal epithelial cells revealed that autophagy-deficient mice exhibited defects in the extracellular secretion of granules in PCs, which led to spontaneous inflammatory bowel disease (IBD) development [73]. For example, Atg16l1 T300A mice mimic human Crohn’s disease features, with fewer secretory granules and disorganized lysozyme [74]. More recent research has further revealed that PCs can regulate autophagy through innate immune receptors such as nucleotide-binding oligomerization domain-containing protein 2 (NOD2), which recognizes commensal bacteria. NOD2 interacts with the vesicle regulatory protein leucine-rich repeat kinase 2 (LRRK2) and receptor-interacting protein kinase 2 (RIPK2) to modulate cellular autophagy, ultimately affecting the secretion of antimicrobial peptides in PCs [75]. Studies show that faulty autophagy disrupts PC secretion, weakening their antimicrobial function and gut balance, which contributes to Crohn’s disease.

Autophagy is vital for the metabolism, proliferation, and differentiation of ISCs into intestinal epithelial cells (IECs) [70]. In ISCs, autophagy reduces harmful reactive oxygen species (ROS) buildup, helping them to function. Without autophagy, the ROS levels rise, damaging ISC activity. Mice lacking Atg5 in their gut cells (Atg5^ΔIEC^) showed fewer ISCs and weaker gut repair after radiation damage. Tests showed higher ROS in Atg5^ΔIEC^ ISCs. Treatment with the antioxidant *N*-acetyl-L-cysteine (NAC) alleviated the damage to ISCs in autophagy-deficient mice [76]. This confirms that autophagy protects ISCs by clearing ROS [77]. Nighot and Zhang observed that autophagy through the lysosomal pathway regulated epithelial barrier function by degrading tight junction protein claudin-2 in Caco-2 cells, a model for IECs. Dysfunction of the autophagic-lysosomal pathway led to elevated claudin-2 levels and increased intestinal permeability [78,79]. Studies in *Drosophila* further confirmed these findings, as defects in autophagy-related genes such as Atg1, Atg13, and Atg17/Fip200 resulted in increased intestinal permeability [80]. Therefore, autophagy defects in IECs lead to compromised intestinal barrier function, making the intestine more susceptible to the absorption of harmful substances into the bloodstream, thereby damaging the organism [81].

#### 4.2.2. Autophagy and Gut Microbiota

The gut microbiota is tightly regulated by the autophagic–lysosomal pathway. Compared to wild-type mice, mice with the specific knockout of Atg5 or Atg7 in colon epithelial cells exhibited dysbiosis, along with a significant increase in the antimicrobial peptide levels [82]. Enhancing autophagy can reduce the levels of mutant huntingtin (mHTT) and promote cell survival in both cellular and animal models of Huntington’s disease (HD) [83]. Another study revealed that gut-specific Atg5 knockout mice exhibited notable alterations in their gut microbiota, including reduced microbial diversity. Pro-inflammatory bacteria (e.g., *Arthrobacter* spp.) and pathogenic bacteria (e.g., *Pasteurella* spp.) were enriched, while anti-inflammatory bacteria (e.g., *Akkermansia muciniphila* and *Bacteroides fragilis*) were decreased, leading to a marked increase in potential pathogens [9]. Autophagy defects in organs like the liver also alter the gut microbiota composition [84]. On the other hand, the gut microbiota has been shown to influence the autophagic activity of intestinal cells. Research indicates that germ-free mice exhibit significantly reduced autophagic activity in their colonic epithelial cells compared to conventionally housed mice. However, the colonization of germ-free mice with *Fibrobacter succinogenes* could restore autophagic activity in the gut epithelial cells, suggesting that the gut microbiota can modulate intestinal autophagy through the production of butyrate [10]. *Lactobacillus* species also boost autophagy by producing indole-3-lactic acid from tryptophan [11].

Environmental factors such as diet, stress, sleep deprivation, and exposure to toxins significantly influence the gut microbiota composition, leading to dysbiosis. This imbalance disrupts the microbiota–gut–brain axis, potentially impairing the autophagy–lysosomal pathway (ALP) and contributing to neurodegenerative diseases [62]. Western diets reduce beneficial microbes like *Bifidobacteria* and *Lactobacilli*, while increasing pro-inflammatory bacteria such as *Enterobacteriaceae* [60]. This reduction leads to decreased production of SCFAs, such as butyrate, which are crucial in maintaining intestinal barrier integrity and modulating inflammation. The compromised barrier allows endotoxins like lipopolysaccharide (LPS) to enter the systemic circulation, triggering systemic inflammation and neuroinflammation, thereby impairing the ALP and promoting neurodegeneration [85]. Chronic stress activates the hypothalamic–pituitary–adrenal (HPA) axis, increasing the cortisol levels, which alters the gut microbiota diversity. This dysbiosis enhances gut permeability, facilitating the translocation of microbial metabolites and pro-inflammatory cytokines into the bloodstream. These substances can cross the blood–brain barrier, activate microglia, and induce neuroinflammation, thereby disrupting the ALP and contributing to neurodegenerative processes [85]. Chronic sleep deprivation alters the gut microbiota composition, increasing NLRP3 inflammasome expression and activating GSK-3β in the hippocampus [86]. These changes disrupt the autophagic flux and lead to tau protein hyperphosphorylation—hallmark features of Alzheimer’s disease. Transplanting the microbiota from sleep-deprived mice into healthy ones replicated these pathological changes, underscoring the role of gut dysbiosis in ALP impairment and neurodegeneration [87]. Exposure to environmental toxins, such as pesticides and heavy metals, can disrupt the gut microbiota composition, leading to dysbiosis [88]. This imbalance affects the production of neuroactive compounds and SCFAs, compromising the gut barrier and facilitating neurotoxicant entry into the central nervous system [89]. The resulting neuroinflammation and oxidative stress can impair the ALP, promoting neurodegeneration.

#### 4.2.3. Autophagy and Gut Immunity

The role of autophagy in regulating intestinal immunity has been extensively studied and recognized. Autophagic defects in IECs disrupt the balance between Th17 and Treg cells, leading to inflammation and autoimmune issues [90]. Dysbiosis-induced autophagy dysfunction also results in excessive ROS production, compromised mucus production, and impaired barrier function, ultimately exacerbating immune responses [91]. In macrophages, the loss or mutation of the autophagy-related gene Atg16l1 leads to increased ROS levels and suppressed cell proliferation. Further studies showed that macrophages from bone marrow lineage-specific (Lysm-Cre) Atg16l1 knockout mice exhibited an increase in cell numbers, elevated pro-inflammatory cytokines, and a higher bacterial load in the intestine, suggesting that autophagy defects exacerbate intestinal inflammation [92]. Moreover, autophagy activation in macrophages overexpressing the autophagy-regulating molecule LRRK2 following *Mycobacterium leprae* infection has been observed. In contrast, the inhibition of LRRK2 reduced TNF-α production in dendritic cells and ameliorated colitis induced by dextran sulfate sodium (DSS) in mice [93]. Autophagy is involved in pathogen degradation, lymphocyte development, antigen presentation, and the release of pro-inflammatory factors to prevent pathogen invasion [94,95]. In epithelial cells, autophagy regulates the secretion of pro-inflammatory factors in response to bacterial infections. In macrophages and dendritic cells, autophagy inhibits the secretion of interleukin (IL)-1β and IL-18 [96].

### 4.3. Related Interventions

Recent clinical and animal model studies have demonstrated that the gut–brain axis plays a crucial role in the pathogenesis and progression of neurodegenerative diseases, suggesting that interventions targeting the gut microbiota, such as probiotic treatments (Figure 3), may be a potential strategy for the prevention and treatment of neurodegenerative disorders [97]. A meta-analysis of clinical randomized controlled trials revealed that probiotics could improve cognitive function in AD patients by reducing inflammation and modulating the redox balance [98]. In AD model mice (TgCRND8), injecting latrepirdine enhanced autophagy, reduced Aβ42 accumulation, and improved behavioral deficits [99]. Additionally, another study showed that long-term latrepirdine treatment in mice increased the expression of autophagy marker LC3 in brain tissue and facilitated the degradation of α-synuclein [100]. Engevik et al. found that treatment with *Bifidobacterium dentium* upregulated the expression of autophagy-related genes in germ-free mice and enhanced intestinal mucus secretion, thereby strengthening the gut mucus barrier [66]. Studies in Atg7-deficient mice and cells showed that *Bifidobacterium breve* (Bb-CM) triggered autophagy in gut cells via MAPK signaling and Atg7 [68]. In a germ-free viral gastroenteritis pig model infected with human rotavirus, treatment with *Lactobacillus rhamnosus GG* alone or in combination with *Lactobacillus reuteri* (ZJ617) induced intestinal cell apoptosis, improved the expression of tight junction proteins related to barrier function, inhibited the activation of the MAPK and NF-κB inflammatory signaling pathways, suppressed intestinal autophagy and oxidative stress, and thus maintained intestinal homeostasis [69,101]. Treating 3xTg-AD mice with SLAB51 (a probiotic mix) activated the NAD+/SIRT1 (Sirtuin-1) pathways, boosted neuronal autophagy, reduced Aβ aggregates, and slowed cognitive decline [67]. Studies have shown that, in addition to influencing the nervous system, probiotic treatments can regulate cardiovascular, renal, and hepatic function through autophagy [102]. This suggests that, beyond the gut–brain axis, autophagy may also play a role in various pathophysiological processes through the gut–organ axis.

## 5. Conclusions

The gut–brain axis, mediated by multiple pathways, including neural, metabolic, and immune signals, facilitates bidirectional regulation between the gut and the brain. Extensive research has demonstrated the critical roles of gut homeostasis and the gut–brain axis in the pathogenesis of neurodegenerative diseases. The autophagic–lysosomal pathway not only protects the nervous system by clearing protein aggregates and maintaining cellular homeostasis, thus counteracting neurodegenerative diseases, but also influences the onset and progression of these diseases by maintaining homeostasis in the peripheral organs, including the gut (Figure 4). Yang and Zhang’s work highlights emerging strategies like aggrephagy-specific modulation and Autophagy-Tethering Compounds (ATTECs), which enable the selective degradation of pathogenic aggregates such as mutant huntingtin (mHTT) [13]. However, the intricate nature of autophagy, the compromised autophagic function within cells affected by neurodegenerative diseases, and the scarcity of viable druggable targets collectively render the clinical application of therapies targeting the ALP to treat neurodegenerative disorders a formidable challenge. Our review contributes to this evolving landscape by integrating gut–brain axis regulation and environmental influences, providing a broader framework for the upstream modulation of the ALP. By linking microbiota dysbiosis, inflammation, and autophagic dysfunction, we propose that targeting both systemic and cellular contributors to ALP disruption may offer more effective and sustainable neuroprotective strategies. This holistic view supports the development of multi-modal therapeutics that combine microbial, dietary, and molecular interventions to restore ALP homeostasis and mitigate neurodegeneration. Future studies investigating the roles and mechanisms of the autophagic–lysosomal pathway in gut homeostasis and the regulation of brain function through the gut–brain axis will provide new insights into the pathogenesis and prevention of neurodegenerative diseases.

## Figures and Tables

**Figure 1 biomedicines-13-01390-f001:**
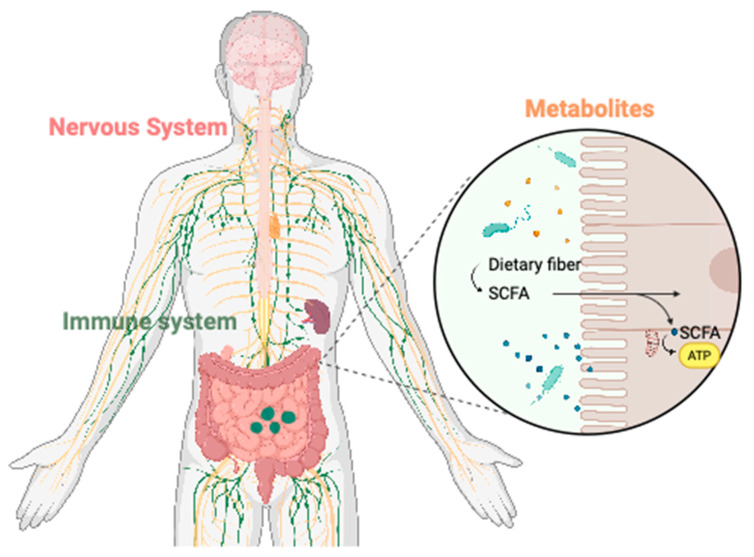
Gut–brain axis mechanisms in neurodegenerative diseases. Created in Biorender. Yao. (2025) https://www.biorender.com/19suj6o. The gut–brain axis (GBA) mediates bidirectional communication between the gut and brain through the nervous system (pink), immune system (green), and microbial metabolites (orange). Gut microbiota ferment dietary fiber to produce short-chain fatty acids (SCFAs), which regulate intestinal barrier integrity, immune responses, and neurotransmission. Signals from the gut are transmitted to the brain via the enteric nervous system (ENS), circulation, and immune pathways, while the central nervous system (CNS) modulates gut function to maintain host homeostasis.

**Figure 2 biomedicines-13-01390-f002:**
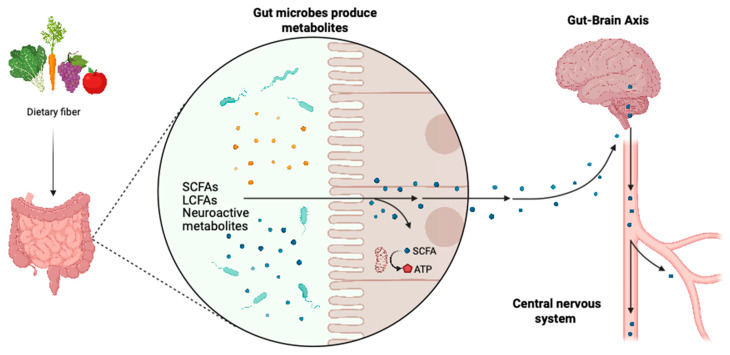
Metabolites produced by the gut microbiota regulate the nervous system through the gut–brain axis. Created in Biorender. Yao. (2025) https://BioRender.com/cnz7i68. Metabolites synthesized by the gut microbiota, including short-chain fatty acids (SCFAs), long-chain fatty acids (LCFAs), and various neurotransmitters, play a crucial role in regulating the nervous system via the gut–brain axis. This bidirectional communication pathway allows these microbial-derived metabolites to cross the intestinal barrier, enter the bloodstream, and ultimately reach the central nervous system, where they modulate neuronal activity, neuroinflammation, and neurotransmission, influencing cognitive function, mood regulation, and various neurological processes.

**Figure 3 biomedicines-13-01390-f003:**
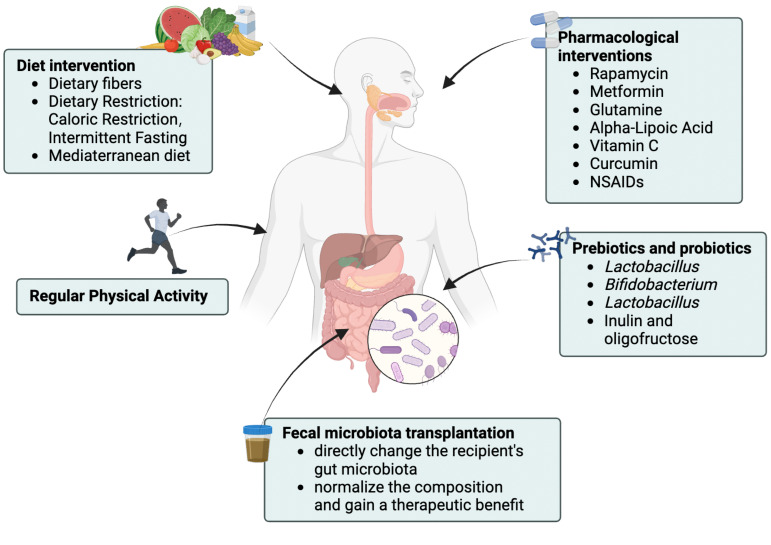
Interventions for neurodegenerative diseases. Created in Biorender. Yao. (2025) https://BioRender.com/ax3wcye. (1) Dietary interventions: High-fiber diets increase short-chain fatty acid production, improving gut barrier integrity, glucose and lipid metabolism, immune regulation, inflammation, and blood pressure [103]. Polyphenols, such as tea polyphenols and grape seed extract, inhibit pathogenic bacteria and promote beneficial microbes [104]. The Mediterranean diet, characterized by high intake of vegetables, fruits, whole grains, and legumes, along with reduced consumption of red meat and processed carbohydrates, modulates the gut microbiota composition to improve vascular and cardiac function. (2) Pharmacological interventions: Rapamycin, a direct mTORC1 inhibitor, induces autophagy, reduces inflammation and immune dysfunction, and restores gut barrier integrity by suppressing mTOR signaling [105,106,107]. Metformin improves metabolic pathways and inflammatory status, increases the abundance of beneficial bacteria such as *Bacteroides* and *Prevotella*, and contributes to gut microbiota remodeling. (3) Probiotics and prebiotics: Probiotics enhance gut microbiota homeostasis by promoting the growth of endogenous beneficial microbes [108]. Prebiotics, including fructo-oligosaccharides and inulin, serve as substrates for beneficial bacteria, promoting their growth and improving gut metabolic function [109]. (4) Exercise: Exercise modulates gut microbiota diversity, specific bacterial populations, and microbial metabolites by reducing systemic inflammation and improving insulin sensitivity, thereby indirectly enhancing gut health [110,111]. (5) Fecal microbiota transplantation (FMT): FMT restores the gut microbial composition, alleviates intestinal inflammation and barrier disruption, reduces systemic inflammation, and improves gastrointestinal dysfunction and motor deficits in PD mice [12].

**Figure 4 biomedicines-13-01390-f004:**
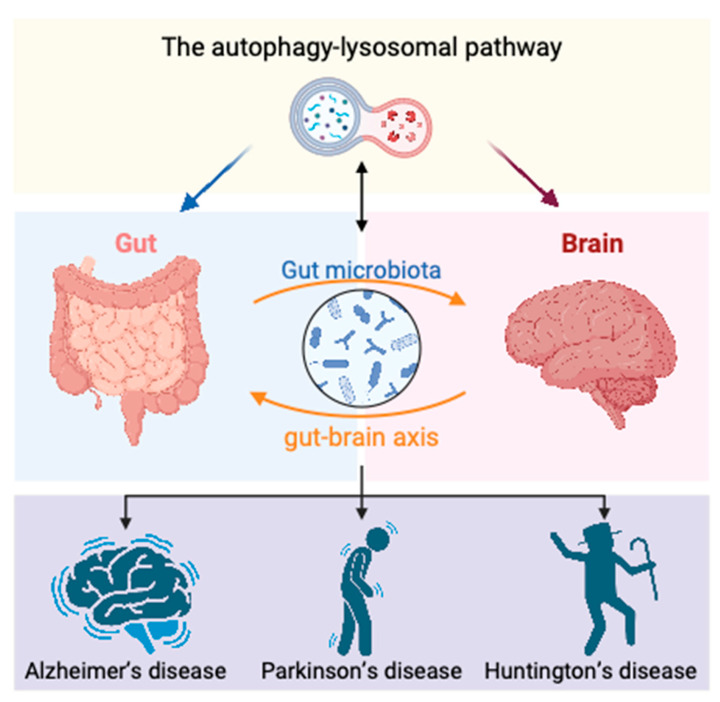
Illustrates the mechanisms by which the autophagy–lysosomal pathway regulates neurodegenerative diseases via the gut–brain axis (GBA). Created in Biorender. Yao. (2025) https://BioRender.com/zdcfjw3. The gut and brain communicate bidirectionally through neural, endocrine, and immune pathways mediated by the GBA. The gut microbiota plays a pivotal role in the pathogenesis of central nervous system (CNS)-related diseases by directly or indirectly modulating GBA function. Autophagy deficiency-induced gut dysbiosis triggers a sustained immune response, promoting the migration and infiltration of immune regulatory cells into the brain, thereby inducing neuroinflammation through microglial activation. The gut microbiota is regulated by autophagy and autophagy-related proteins, while also influencing cellular autophagy under physiological conditions. The complex bidirectional interaction between the gut microbiota and autophagy provides novel insights into the pathogenesis and progression of neurodegenerative diseases.

**Table 1 biomedicines-13-01390-t001:** Summary of studies on ALP in neurodegeneration via gut–brain axis.

Study	Disease Model	Experimental Type	ALP Intervention/ Target	Outcome Summary
Kim et al., 2024 [59]	APP/PS1 mice	In vivo	Modulation of astrocytic autophagy	Enhanced Aβ clearance and improved cognitive function through astrocytic autophagy plasticity.
Zeng et al., 2025 [58]	Astrocyte cultures	In vitro	Restoration of lysosomal acidification	Impaired lysosomal acidification in astrocytes contributes to neuroinflammation; restoration ameliorates inflammatory responses.
Mitra et al., 2023 [62]	Various neurodegenerative models	Review of in vitro and in vivo studies	Gut microbiota modulation	Gut microbiota influences autophagy regulation; potential therapeutic avenue in neurodegeneration.
Luan et al., 2023 [63]	Human samples and cell lines	In vitro	Interaction of bile acids with γ-secretase	Microbiota-derived bile acids promote γ-secretase activity via Nicastrin, increasing Aβ production.
Jiao et al., 2024 [64]	PD models (cellular and animal)	In vitro and in vivo	Targeting ALP via chemical and gene therapy	Upregulation of ALP facilitates clearance of α-synuclein aggregates; potential therapeutic strategy in PD.
Tunold et al., 2024 [65]	PD patient cohorts	Clinical study	Analysis of lysosomal polygenic burden	Higher lysosomal polygenic scores associated with accelerated cognitive decline in PD patients with low AD risk.
Yang et al., 2023 [13]	HD models	Review of in vitro and in vivo studies	Pharmacological targeting of ALP	Enhancing ALP activity reduces mutant huntingtin aggregates; promising therapeutic approach in HD.
Kim et al., 2020 [17]	ADLPAPT mice (AD model)	In vivo	Fecal microbiota transplantation (FMT)	FMT from healthy donors reduced Aβ plaques and tau pathology and improved cognition.
Chen et al., 2022 [16]	3xTg-AD mice (germ-free vs. SPF)	In vivo	FMT from AD vs. healthy donors	FMT from AD donors aggravated AD pathology; GF status mitigated symptoms.
Dodiya et al., 2022 [18]	APP/PS1 mice	In vivo	Antibiotic-induced microbiota depletion	Antibiotic treatment reduced Aβ deposition.
Sampson et al., 2016 [7]	Thy1-αSyn mice (PD model)	In vivo	FMT from PD vs. healthy subjects	PD-derived microbiota worsened α-syn pathology and motor symptoms.
Singh et al., 2023 [22]	α-Syn overexpressing rats	In vivo	Aging + gut microbiome analysis	Gut dysbiosis correlated with α-syn aggregation and inflammation.
Sun et al., 2018 [23]	MPTP-induced PD mice	In vivo	FMT + TLR4/TNF-α pathway analysis	Healthy FMT alleviated motor deficits, reduced neuroinflammation.
Bhattarai et al., 2021 [24]	Rotenone PD model	In vivo (germ-free vs. SPF)	Gut microbiota manipulation	Motor and GI symptoms only induced in SPF mice, not GF.
Kong et al., 2020 [29]	R6/1 HD mice	In vivo	Gut microbiome profiling	HD mice had increased Bacteroidetes, decreased Firmicutes; gut dysbiosis linked to weight loss and behavior.
Stan et al., 2020 [30]	R6/2 HD mice	In vivo	Intestinal barrier markers	Observed increased permeability, reduced body size, worsened gut integrity.
Engevik et al., 2019 [66]	Germ-free mice	In vivo	*Bifidobacterium dentium* treatment	Stimulated autophagy gene expression and enhanced mucus secretion.
Bonfili et al., 2018 [67]	3xTg-AD mice	In vivo	SLAB51 probiotic mix	Activated SIRT1 pathway, induced neuronal autophagy, reduced Aβ burden.
Inaba et al., 2016 [68]	Atg7-deficient gut cells	In vitro	*B. breve* culture medium	Induced autophagy via MAPK pathway, restored gut epithelial function.
Cui et al., 2017 [69]	Mouse intestinal cells	In vitro and in vivo	*L. reuteri* ZJ617	Improved tight junctions; reduced autophagy dysfunction and inflammation.

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
