# Peer review of "Advances in Autophagy–Lysosomal Pathway and Neurodegeneration via Brain–Gut Axis"

_biomedicines, 2025, doi:10.3390/biomedicines13061390_

Round 1
Reviewer 1 Report
Comments and Suggestions for Authors
The manuscript titled "Advances in Autophagy-Lysosomal Pathway and Neurodegeneration via Brain-Gut Axis " aimed to "systematically collect and analyze relevant literature published in both domestic and international studies (copied statement from the abstract: "Methods:
Relevant literature published over the past decade was systematically collected and analyzed from both domestic and international studies.").
Major concern:
Unfortunately, this is by no means a systematic review. PRISMA recommendations should be used to adequately accomplish the intended aim of the study. Without following PRISMA rules and guidelines, this work cannot be accepted.
In addition, instructions for authors have not been carefully followed.
Technical errors were noticed.
The abstract is inappropriate, too long and not written according to instructions for authors.
The
Comments on the Quality of English LanguageEnglish language and style needs minor polishing
Author Response
Comments 1: Unfortunately, this is by no means a systematic review. PRISMA recommendations should be used to adequately accomplish the intended aim of the study. Without following PRISMA rules and guidelines, this work cannot be accepted.
Response 1: It is greatly appreciated that you took the time to review our work. We regret any confusion caused by the original manuscript’s mistake of review type declaration and the errors in the abstract pointed out by you have been corrected. We would like to clarify that this review was designed as a narrative/scoping review rather than a systematic review or meta-analysis. The primary aim was to synthesize emerging interdisciplinary evidence on the autophagy-lysosomal pathway (ALP) and gut-brain axis interactions in neurodegeneration, with an emphasis on mechanistic insights, translational challenges, and future directions.
The revised methods are as follows: This narrative review synthesizes evidence from preclinical, clinical, and translational studies (2014–2025) to explore the interplay between the autophagy-lysosomal pathway (ALP) and gut-brain axis in neurodegeneration. Literature was identified via PubMed and Web of Science us-ing search terms including autophagy, lysosome, gut microbiota, neurodegeneration, and gut-brain axis, with additional manual screening of reference lists. Inclusion criteria prioritized studies elu-cidating molecular mechanisms (e.g., ALP-microbiota crosstalk), while excluding case reports or non-peer-reviewed sources.
Comments 2: In addition, instructions for authors have not been carefully followed.Technical errors were noticed. The abstract is inappropriate, too long and not written according to instructions for authors.
Response 2: Thank you for pointing this out. We agree with this comment. Therefore, we have revised the abstract according to the instructions for author as below(Page 1).
Abstract: Background/Objectives: The autophagy-lysosomal pathway (ALP) is crucial for neuronal health by clearing misfolded proteins and damaged organelles. While much research has focused on ALP dysfunction in the central nervous system, new evidence shows its importance in the gut, where it affects neurodegeneration via the gut-brain axis. Past reviews have mainly studied ALP's direct neuroprotective effects or the gut microbiota's role in neurodegeneration separately. However, the two-way relationship between ALP and the gut microbiota in neurodegenerative diseases is not well understood. We combine the latest findings on ALP's role in gut health, microbial imbalance, and neuroinflammation, providing a comprehensive view of their combined effects in Alzheimer’s, Parkinson’s, and Huntington’s diseases. Methods: See above. Results: The gut-brain axis facilitates bidirectional communication between the gut and the brain through neural, immune, and metabolic pathways. Autophagy dysfunction may disrupt intestinal homeostasis, promote gut microbiota dysbiosis, and trigger chronic neuroinflammation, ultimately accelerating neurodegeneration. Notably, strategies targeting gut microbiota and restoring intestinal barrier function via the ALP have demonstrated promising potential for delaying the progression of neurodegenerative diseases. Conclusions: This review establishes ALP as a dynamic regulator of gut-brain communication, highlighting microbiota-targeted therapies as promising strategies for neurodegeneration.
Comments 3: English language and style needs minor polishing
Response 3: We have polished the language as you suggested. Since you mentioned that the English language and style only needed minor polishing, we've carefully reviewed and refined the text to enhance its readability and overall quality. Now, the language in the manuscript is more polished and adheres to academic writing standards.
Reviewer 2 Report
Comments and Suggestions for Authors
The manuscript by Hanlong Han et al., entitled “Advances in Autophagy-Lysosomal Pathway and Neurodegeneration via Brain-Gut Axis” provides a comprehensive and extensive review on the role of autophagy in maintaining homeostasis and mechanistic insights into gut-brain mediated regulation in neurodegenerative diseases. The manuscript is well written with detailed information about the role of autophagy in various neuro degenerative diseases. Including some minor changes mentioned below will further improve the manuscript:
- Kindly include a schematic figure illustrating how metabolites affect neurodegenerative diseases through the various pathways discussed in Section 3.2.
- Expand the section-4.2.2 (Autophagy and Gut Microbiota). Currently, the section is short. Expand more on various pathogenic and non-pathogenic bacteria involved via autophagic lysosomal pathway.
Author Response
Comments 1: Kindly include a schematic figure illustrating how metabolites affect neurodegenerative diseases through the various pathways discussed in Section 3.2.
Response 1: It is a great help that you provided such valuable insights in your review. Therefore, we have included a schematic figure titled "Figure 2. Metabolites Produced by Gut Microbiota Regulate the Nervous System Through the Brain-Gut Axis" on pages 5 - 6. Your guidance will be highly valued.
Comments 2: Expand the section 4.2.2 (Autophagy and Gut Microbiota). Currently, the section is short. Expand more on various pathogenic and non-pathogenic bacteria involved via the autophagic lysosomal pathway.
Response 2: Agree.Thank you for pointing this out. Accordingly, we have further elaborated on how the autophagic-lysosomal pathway is involved in gut microbiota dysbiosis to underscore this critical aspect. Specifically, as detailed on pages 9 - 10, Western diets reduce beneficial microbes like Bifidobacteria and Lactobacilli, while increasing pro-inflammatory bacteria such as Enterobacteriaceae[57]. This reduction leads to decreased production of SCFAs, such as butyrate, which are crucial for maintaining intestinal barrier integrity and modulating inflammation. The compromised barrier allows endotoxins like lipopolysaccharides (LPS) to enter systemic circulation, triggering systemic inflammation and neuroinflammation, thereby impairing ALP and promoting neurodegeneration[88].
Reviewer 3 Report
Comments and Suggestions for Authors
Date: 30th April 2025
Journal: Biomedicines (ISSN 2227-9059)
Manuscript ID: biomedicines-3604445
Type: Review
Title: Advances in Autophagy-Lysosomal Pathway and Neurodegeneration via Brain-Gut Axis.
Authors: Ping Yao, Hanlong Han*.
Brief Summary of the article
The article by Yao et.al., and team discusses the role of Autophagy-Lysosomal Pathway and Neurodegeneration via Brain-Gut Axis. The review is well designed and could be crucial in identifying novel molecular mechanism and targets related to ALP in Neurodegeneration. I have a few queries and suggestions that could be included in the revised manuscript to improve the overall quality and content of the article.
- The similarity Index (Percent match: 35%) of the review is extremely high. Please reduce the same to below 15%.
- Introduction should be revised and background information on the past literature related to the role of ALP in neurodegenerative diseases via the Brain-Gut Axis should be included. Similarly, sufficient novelty should be provided as to how the current review differentiates from the past literature.
- Please prepare a table with all the past studies conducted in in-vitro and in-vivo experimental models of neurodegeneration linked to ALP mechanism and briefly write the summary and results in the table. This could provide sufficient information on the past literature that could help in designing the future studies.
- Please discuss in brief how environmental factors contribute the microbiome dysbiosis and alter the microbiome-gut-brain axis which could potentially inhibit the ALP and cause neurodegeneration.
- 1 Autophagy-Lysosomal Pathway should be discussed in more detail and recent studies should be included in the discussion.
- Future directions should be included. How do the authors feel this review could help in identifying novel therapeutics targeting ALP in neurodegeneration? Please include the same.
- References should be modified and recent studies from 2023-2025 should also be included in the review.
Author Response
Comments 1: The similarity Index (Percent match: 35%) of the review is extremely high. Please reduce the same to below 15%.
Response 1: It is kind of you to give such a detailed review. We have reduced the similarity rate to below 15%.
Comments 2: Introduction should be revised and background information on the past literature related to the role of ALP in neurodegenerative diseases via the Brain-Gut Axis should be included. Similarly, sufficient novelty should be provided as to how the current review differentiates from the past literature.
Response 2: Thank you for pointing this out. We agree with this comment. Therefore, we have revised the introduction as you requested. Specifically, as detailed on pages 1 2:
Historically, research has focused on central nervous system (CNS)-centric mechanisms, such as amyloid-β (Aβ) plaques in AD and α-synuclein (α-syn) aggregation in PD. However, emerging evidence highlights the gut-brain axis (GBA) as a critical mediator of neurodegeneration, with bidirectional communication involving neural, immune, and metabolic pathways[3].
While previous reviews have established ALP's role in neuronal homeostasis, recent studies reveal its regulatory influence on peripheral organs, particularly the gut. For instance, intestinal epithelial cell-specific Atg5 or Atg7 knockout disrupts gut microbiota balance, enriching pro-inflammatory bacteria (e.g., Pasteurella) and exacerbating systemic inflammation via disrupted barrier integrity[9]. Conversely, gut microbiota-derived metabolites, such as butyrate and indole-3-lactic acid, enhance ALP activity through AMPK/mTOR and AhR-TFEB pathways, respectively[10, 11].
By bridging the mechanistic insights of autophagy regulation with the emerging concept of the GBA, this review provides a comprehensive perspective on how targeting the ALP—both centrally and peripherally—could offer novel therapeutic approaches for NDDs. It further highlights translational innovations, such as fecal microbiota trans-plantation (FMT) and drug discovery via autophagy tethering, which target ALP dysregulation across organs[12, 13]. These advancements not only refine existing hypotheses but also pave the way for multi-modal therapies addressing the systemic nature of neurodegeneration.
Comments 3: Please prepare a table with all the past studies conducted in in-vitro and in-vivo experimental models of neurodegeneration linked to ALP mechanism and briefly write the summary and results in the table. This could provide sufficient information on the past literature that could help in designing the future studies.
Response 3: Agree. We have, accordingly, prepared a table with all the past studies conducted in in-vitro and in-vivo experimental models of neurodegeneration linked to ALP mechanism and briefly written the summary and results in Table 1. (Page 7-8)
Table 1. Summary of Studies on ALP in Neurodegeneration via Gut-Brain Axis
Comments 4: Please discuss in brief how environmental factors contribute the microbiome dysbiosis and alter the microbiome-gut-brain axis which could potentially inhibit the ALP and cause neurodegeneration.
Response 4: Thank you very much for your valuable comments. We have added the discussion on how environmental factors contribute to the dysbiosis of the microbial community in 4.2.2. Autophagy and Gut Microbiota (Pages 9 10) of the manuscript.
Comments 5: Autophagy-Lysosomal Pathway should be discussed in more detail and recent studies should be included in the discussion.
Response 5: Thank you again for your valuable comments. We have discussed the autophagy lysosomal pathway in detail and incorporated the discussion in light of recent research in 4.2.2. Autophagy and Gut Microbiota (Pages 9-10).
Comments 6: Future directions should be included. How do the authors feel this review could help in identifying novel therapeutics targeting ALP in neurodegeneration? Please include the same.
Response 6: Thank you for pointing this out. In order to respond to your review, we are about to add the discussion on future directions to the conclusion (Page 11-12) as follows:
Yang and Zhang’s work highlights emerging strategies like aggrephagy-specific modulation and AuTophagy-TEthering Compounds (ATTECs), which enable selective degradation of pathogenic aggregates such as mutant huntingtin (mHTT)[13]. However, the intricate nature of autophagy, the compromised autophagic function within cells affected by neurodegenerative diseases, and the scarcity of viable druggable targets collectively render the clinical application of therapies targeting the autoph ALP for treating neurodegenerative disorders a formidable challenge.
Our review contributes to this evolving landscape by integrating gut-brain axis regulation and environmental influences, providing a broader framework for upstream modulation of ALP.
By linking microbiota dysbiosis, inflammation, and autophagic dysfunction, we propose that targeting both systemic and cellular contributors to ALP disruption may offer more effective and sustainable neuroprotective strategies. This holistic view supports the development of multi-modal therapeutics that combine microbial, dietary, and molecular interventions to restore ALP homeostasis and mitigate neurodegeneration.
Future studies investigating the roles and mechanisms of the autophagic-lysosomal pathway in gut homeostasis and the regulation of brain function through the gut-brain axis will provide new insights into the pathogenesis and prevention of neurodegenerative diseases.
Comments 7: References should be modified and recent studies from 2023-2025 should also be included in the review.
Response 7: In accordance with the above revision suggestions, the latest research from 2023 to 2025 has been incorporated into the review. We warmly welcome you to check it.
Round 2
Reviewer 1 Report
Comments and Suggestions for Authors
Authors made improvements to the manuscript.
Comments on the Quality of English LanguageEnglish language needs some improvement.